# Small Intestinal Atresia: Should We Preserve the Peel or Toss It?

**DOI:** 10.3390/children12020240

**Published:** 2025-02-17

**Authors:** Benedetta Marino, Giulia Mottadelli, Marta Bisol, Maria Sergio, Piergiorgio Gamba, Elisa Zambaiti

**Affiliations:** 1UOC Pediatric Surgery, Department of Women’s and Children’s Health, University of Padua, 35128, Padua, Italy; benedetta.marino22@gmail.com (B.M.); piergiorgio.gamba@unipd.it (P.G.); 2UOC Pediatric Surgery, Ospedale Infantile “Cesare Arrigo”, 15121 Alessandria, Italy; gmottadelli@ospedale.al.it; 3UOC Pediatric Surgery, Azienda Ospedaliera Universitaria “Paolo Giaccone”, 90127 Palermo, Italy; maria.sergio@unipa.it; 4UOC Pediatric Surgery, Ospedale Infantile Regina Margherita, 10126 Torino, Italy

**Keywords:** intestinal atresia, short bowel syndrome, neonate

## Abstract

Background: Apple peel atresia (APA) is a rare and severe form of intestinal atresia, but little is known on long-term outcomes. We compared outcomes of apple-peel atresia based on different surgical approaches. Methods: a retrospective review from two institutions compared APA-resected and APA-preserved patients. Demographics, operative details, postoperative courses and long-term outcomes were analyzed. Results: Of the 16 APA neonates, 10 (62.5%) were in APA-resected and 6 (37.5%) in APA-preserved groups. Early postoperative complications occurred in 7 patients (43.75%) including vomiting, infection, intestinal occlusion, anastomotic dehiscence, multiorgan failure, equally distributed among groups. Length-of-stay is higher in the APA-preserved group (median 67 vs. 27 days, *p* = 0.14). Overall survival at discharge was 87.5%. Twelve children (75%) were followed for an average of 5 years. Reoperation was required in 4 children owing to anastomotic obstruction and adhesive intestinal obstruction, two in each group. Conclusion: to prevent intestinal failure, keeping the APA shows not inferior results compared to resection, even though it may have a longer first-postoperative course.

## 1. Introduction

Intestinal atresia is a very rare congenital anomaly due to an embryological defect of the intestinal vascular development [1]. The total prevalence of small intestinal atresias, in particular, jejunal-ileal atresia, is not well known and in literature it is described a varied prevalence from 1 in 330 to 0.8 per 10,000 live births in Europe [2].

Prenatal diagnosis is possible, even if difficult, with the use of prenatal ultrasound. A suspicion is based on the identification of unspecific signs, among which the most common is the detection of small intestine dilation, which typically appears in late gestation (primarily after 29 weeks). Unfortunately, nearly half of small bowel obstruction cases are not diagnosed during pregnancy and therefore they constitute a novel and often unpleasant finding at birth [3].

In these kinds of patients it is frequent the association with preterm birth, low birth weight and associated malformations, that are known as predicting factors of increased morbidity [4]. The risk of mortality remains high, especially in developing countries, because of lack of prenatal diagnosis, difficult access to tertiary referral hospitals and high rates of infections [5].

Apple-peel atresia (APA) is a rare and serious variant that occurs in less than 10% of all small intestinal atresia [6], in which the proximal bowel is dilated and the distal tract is wind up on a branch of ileocolic artery. Previous studies have revealed that proximal jejunal atresia (less than 20 cm from Treitz ligament), type IIIb atresia (apple-peel atresia), type IV atresia (multiple atresia), and complicated intestinal atresia by volvulus or meconium peritonitis are the most severe variant, associated with poor outcomes such as mean duration of parenteral nutrition, oral feed tolerance, secondary surgery and mortality [7]. Difficult recovery is principally due to residual small intestine length [8,9], that can be lacking in APA patients because of the sort of the atretic segment.

Various surgical strategies were described to manage severe intestinal atresia [10], but we know little about long term outcomes, specifically for apple-peel atresia [11], in which the balance between resecting an atretic segment opposes the potential benefit of keeping adequate length to ensure intestinal sufficiency.

The purpose of this study was to investigate the postoperative course of apple-peel atresia and long-term outcomes based on different surgical approaches.

## 2. Methods

This was a double center study considering all patients treated for intestinal atresias at Azienda Ospedaliera Universitaria “Paolo Giaccone” of Palermo and Azienda Ospedale-Università di Padova between 2001 and 2018. Of those, only patients with APA were included in the present study. No exclusion criteria were applied as withdrawal from the follow-up was also considered as an outcome.

Diagnosis of APA was based on operative findings.

We collected data from patient characteristics including sex, age and weight at birth, associated malformations. Operative details, postoperative courses such as mean hospital stay, duration of total parenteral nutrition, surgery complication and necessity of secondary intervention were analyzed. Complications were ranked according to the modified Clavien–Dindo classification for surgical complications and considered major if of grade 3 or higher [12].

To describe long-term outcomes we choose to consider mortality rate, reoperation or readmission, sign of malabsorption and growth retardation in a 5 years follow-up.

We then classified the APA neonates in two groups: APA-resected and APA-preserved according to the atretic segment fate, and we compared the outcomes of the two groups.

Patients’ data were presented with descriptive statistics including frequencies, percentages, medians and ranges (as data were considered not normally distributed). Statistical analysis was conducted using GraphPad Prism 8.3.0, San Diego, CA, USA. *p*-value < 0.05 was considered statistically significant. The study was conducted in accordance with the Helsinki Declaration and local regulations. Due to the retrospective design of the study, the project was notified to the ethics board of both centers and required no formal ethical approval. Informed consensus was obtained for all participants included in the study.

## 3. Results

### 3.1. Population Characteristics

During the study period, 16 neonates were treated for APA, of these 10 were female and 6 were male. Two females were twins, born at 32 gestational weeks (GW) from a monochorionic pregnancy. Of the total, 9 patients (56%) were born preterm with an overall median GW at birth of 35 (SD 2) in the entire population. Median weight at birth was 2510 g (SD 659 g).

Prenatal diagnosis was available in 8 (50%) of patients, with 6 identifications of intestinal dilatation, one polyhydramnios and one double bubble sign.

At birth, associated anomalies were diagnosed in 4 patients (25%); among those, two cardiopathies, one congenital choanal stenosis and one presacral dimple. One patient had a heterozygous mutation of CFTR delta F508/N1303K gene identified at genetic testing.

### 3.2. Surgical Approach

All patients underwent surgery at a median of 2 days (SD 1, range 0 to 5). All of them were positioned supine and a transverse laparotomy was performed to deal with the neonatal intestinal obstruction. After the access to the peritoneal cavity, the bowel was explored to determine the level of atresia. Four patients had an associated intestinal malrotation (25%), two presented with a volvolus (12.5%) and one had a bowel perforation (6.25%) at first surgical exploration. Eight patients had a stoma created during the first procedure (50%). Ten patients (62.5%) underwent resection of the APA and primary anastomosis (APA-resected group) while 6 (37.5%) had the APA preserved at initial surgical approach (APA-preserved group). Overall residual bowel length following surgery was 104 cm (SD 38), with a median of 95 cm (SD 41) in the APA-resected and 120 cm (SD 24) in the APA-preserved (Figure 1).

### 3.3. Short Term Outcomes

Early postoperative complications occurred in 7 patients (43.75%) of which 4 were classified as major complications according to Clavien Dindo classification (Figure 2): they included vomiting (1), infection requiring systemic antibiotics either with/without association to central venous catheter infection (3), intestinal obstruction (2) and anastomotic dehiscence (1) requiring emergent reoperation, multi organ failure (1).

Oral feed was started at a median of 10 post-operatives days (SD 3). Median intensive care unit (ICU) and surgical ward hospitalization was of 13 days (SD 26) and 39 (SD 117) respectively. Excluding planned stoma closure, reoperation was required in 4 children (25%), 2 in the APA-preserved group (33.3%) and 2 in the APA-resected group (20%), mainly due to anastomotic stenosis and adhesive intestinal obstruction. Survival at hospital discharge was 87.5%.

### 3.4. Long Term Outcomes

Of the long-term survivors, 4 were lost at follow-up (25%) while 12 children (75%) were followed up for an average of 5 years, of which 7 children (43.7%) showed normal growth and development and 3 were still on partial parenteral nutrition (PN) at the time of the study (18.75%).

### 3.5. Comparison APA-Resected vs. APA-Preserved

APA-resected and APA-preserved groups have similar birth weight (median 2215 vs. 2700 g, *p* = 0.63 *t*-test), gestational age (median 37 vs. 34 GW, *p* = 0.33, *t*-test).

Post-operative days at ICU are higher in the APA-preserved group (median 36 days, SD 41) than the APA-resected group (median 13 days, SD 24), although not significant (*p* = 0.55, *t*-test); also overall length-of-stay is higher in APA-preserved group (median 67 days, SD 156) than the APA-resected group (median 27 days, SD 18), *p* value 0.14, *t*-test. There was one demise in each group. Long term outcomes were also similar between the two groups, with a similar proportion of patients requiring parental nutrition (2 vs. 1 patient in APA resected vs. preserved groups, 20 vs. 17%) although days on PN are lower in the patient APA-preserved (Table 1).

## 4. Discussion

Apple-peel atresia is one of the most complicated forms of intestinal atresia and due to its rareness, surgical treatment is not fully standardized. Different surgical interventions are described for apple-peel atresia treatment in literature, such as primary end-to-end or end-to-side anastomosis associated with tapering [13] or lengthening of dilated bowel [14,15], Santulli stoma with resection of atretic segment [16], T-tube enterostomy [17]. Open surgery is more frequent, but a laparoscopic approach can be used according to team preferences and expertise [18].

Despite multiple surgical strategies, nowadays, it is clear that preserving intestinal length to avoid short bowel syndrome (SBS), in this kind of patients, should be considered the main target [6].

In fact, SBS may be the most alarming consequence of surgery in APA patients, potentially leading to life-long need of parenteral nutrition, which may result in cholestasis and liver dysfunction, eventually associated with central-line infections and venous insufficiency. In literature SBS was defined in different ways considering the length of residual small bowel, preservation of ileocecal valve and laboratory tests. As a general practical rule, we talk about SBS in presence of residual small intestinal length of less than 25% of total bowel length, corrected for age and need of parenteral nutrition more than 6 weeks after surgery [19].

Another important factor that can predict an earlier initiation and tolerance of oral nutrition is the discrepancy in the diameters of the proximal and distal segments of the anastomosis. A bigger discrepancy is associated with longer recovery times because of slower restarting of peristalsis, which can also lead to a higher risk of intestinal fluids stasis and bacterial translocation with sepsis [15].

So far, there is no consistent literature on the preservation of the atretic segment. In fact many surgeons rejected this possibility because of the potential complications due to the maintenance of a dysembryogenetic tissue and the fear of increasing risk of volvulus. To avoid this latter complication, a partial resection associated with mesopexy seems the best surgical treatment [20].

However, atretic segments often involve the majority, if not all, the small intestine and a resection could easily lead to SBS. We therefore choose to evaluate long-term outcomes in APA patients considering factors that can worsen enteral autonomy. Our retrospective review, compared two groups (APA-preserved and APA-resected group) created by different surgical strategies on the management of the atretic segment. The groups were similar in premature birth occurrence and weight at birth, and there was a general prevalence of females, as reported elsewhere in literature. Prenatal diagnosis was made in half of patients by the detection of unspecific signs such as small intestinal dilation, polyhydramnios and double-bubble sign through the use of ultrasounds. Once birthed, it was discovered that 25% of them had associated malformation. In line with the literature, the most frequent were association with intestinal malrotation and cardiopathies [21].

Moreover, as described in previous studies, APA patients frequently deal with a hard and prolonged postoperative course [22]. In our population almost half of the patients had surgical complications including vomiting, infection requiring systemic antibiotics, intestinal obstruction, anastomotic dehiscence and one multi organ failure leading to death. There were no differences in the two groups in terms of surgical complications that need re-interventions, but we found out that the APA-preserved group had higher length-of-stay at the Intensive Care Unit in the postoperative days and longer recovery. These findings are probably due to the difficult resumption of peristalsis of the atretic segment. However, we didn’t notice any difference either in initiation of oral enteral nutrition or in the possibility to achieve enteral autonomy thereafter which was similar in both groups. These results are in line with the literature in which it is suggested that parenteral nutrition may be required as an initial management option, as well as temporary stoma, and early morbidity is considered common [17].

Excluding planned stoma closure, reoperation was required in an equally distributed number of children, due to anastomotic stenosis, suture dehiscence or adhesive intestinal obstruction.

Despite initial morbidity may be high, available literature supports similarly to our report that overall long-term outcomes in APA patients are optimal; at an average of 5 years of follow-up, APA patients show normal intestinal functioning and have no symptoms of failure to thrive with a good percentile of growth [23].

In our cases, survival at hospital discharge was high and deceased were also similar in the two groups. One of them had an interventricular defect and struggled with multi-organ failure after surgery. We cannot correlate these deaths to the preference of a specific surgical approach.

This study was conducted with evident limits such as the poor number of patients, the long inclusion period, and the retrospective collection of data. However, it gives precious insight into the possibility to maintain the atretic segment. The management of APA requires a careful balance between necessary resection of the atretic segment and the preservation of as much functional bowel length as possible. A thorough pre-operative assessment and a tailored surgical approach are critical in minimizing the risk of short bowel syndrome.

## 5. Conclusions

We reported that APA-patients have excellent long-term outcomes overall. In the APA-preserved group, early postoperative courses may be arduous but there seems not to be an increased risk of intestinal complications thereafter. Intestinal length remains the principal prognostic factor for earlier enteral autonomy, which suggests to avoid the resection of atretic segment whenever possible. In order to predict mortality, associated malformations and preterm delivery are also important factors to consider.

## Figures and Tables

**Figure 1 children-12-00240-f001:**
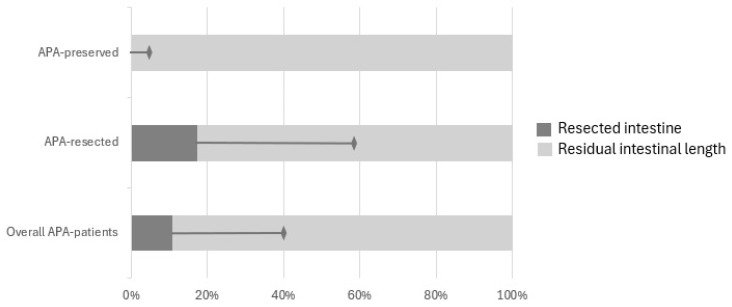
Comparison of resected and residual intestinal length in APA-resected, APA-preserved and overall population. APA = Apple peel atresia.

**Figure 2 children-12-00240-f002:**
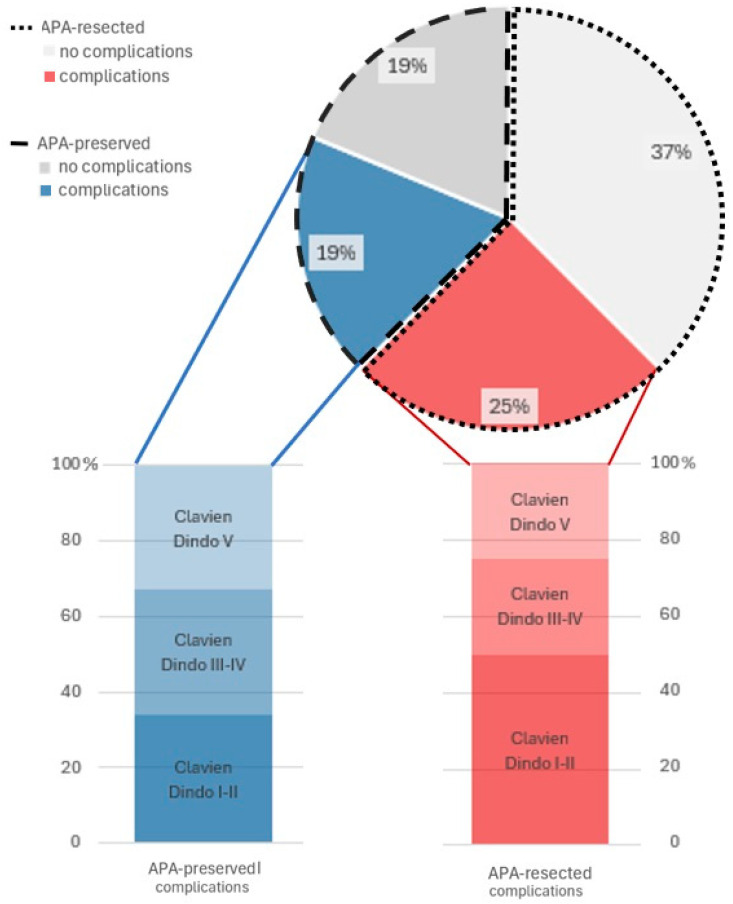
Complication rate in overall population and Clavien-Dindo complications classification distinction between the two groups. APA = Apple peel atresia.

**Table 1 children-12-00240-t001:** Comparison id APA-resected and APA-preserved groups. APA = Apple peel atresia; M = male; F = female; stdev = standard deviaition, n = number, ICU = intensive care unit; PN = parenteral nutrition.

	Overall APA-Patient	APA-Resected	APA-Preserved	*p* Value
	16	10	6	
Gender (M/F)	6/10	2/8	4/2	0.11
Gestational weeks at birth (median, stdev)	35 (3)	34 (3)	37 (3)	0.33
Birth weight in g (median, stdev)	2510 (659)	2215 (662)	2700 (712)	0.63
Malrotation (n, %)	4 (25%)	3 (30%)	1 (17%)	1.00
Stoma formation (n, %)	8 (50%)	6 (60%)	2 (34%)	0.37
ICU length-of-stay, days (median, stdev)	13 (26)	13 (24)	36 (41)	0.55
Oral feed, days (median, stdev)	10 (3)	10 (4)	10 (180)	1.00
Overall length-of-stay, days (median, stdev)	38 (117)	27 (18)	67 (156)	0.14
Long term need for PN (n, %)	3 (19%)	2 (20%)	1 (17%)	1.00
Long-term re-operation (n, %)	4 (25%)	2 (20%)	2 (34%)	0.60

## Data Availability

The data presented in this study are available on request from the corresponding author. The data are not publicly available due to privacy or ethical.

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
