# Peer review of "Small Intestinal Atresia: Should We Preserve the Peel or Toss It?"

_children, 2025, doi:10.3390/children12020240_

Round 1
Reviewer 1 Report
Comments and Suggestions for Authors
Marino et al performed a retrospective review of 16 Apple peel atresia (APA) cases from two institutions in Italy between 2001 and 20018. All cases were divided into APA-resected and APA-preserved groups. The short-term and long-term outcomes were compared. The authors concluded that preserving APA bowel tissue is equally good as resection, if not better, although the post-operative ICU days are longer in APA-preserved group. The manuscript is well-organized and written. Critical comments are listed below:
1. Tables, pie charts and bar graphs should be used for demographics and the comparison of short-term and long-term outcomes in APA-resected and APA-preserved groups. This is to improve clarity and readability.
2. In “Discussion”, this statement “we talk about SBS in presence of less than 25%....length” needs clarification. Does it mean that SBS represents less than 25% of residual length?
3. Please discuss and compare this manuscript to the previously published Ref 25 and 26.
Author Response
Marino et al performed a retrospective review of 16 Apple peel atresia (APA) cases from two institutions in Italy between 2001 and 20018. All cases were divided into APA-resected and APA-preserved groups. The short-term and long-term outcomes were compared. The authors concluded that preserving APA bowel tissue is equally good as resection, if not better, although the post-operative ICU days are longer in APA-preserved group. The manuscript is well-organized and written.
Thank you for your comment
Critical comments are listed below:
1. Tables, pie charts and bar graphs should be used for demographics and the comparison of short-term and long-term outcomes in APA-resected and APA-preserved groups. This is to improve clarity and readability.
We thank you for this suggestion, we updated the tables accordingly.
2. In “Discussion”, this statement “we talk about SBS in presence of less than 25%....length” needs clarification. Does it mean that SBS represents less than 25% of residual length?
Your osservation is correct, the paper has been modified accordingly.
3.Please discuss and compare this manuscript to the previously published Ref 25 and 26.
We added the information required.
Reviewer 2 Report
Comments and Suggestions for Authors
Dear Authors,
Thank you for your paper entitled " Small intestinal atresia: should we preserve the peel or toss it?". The manuscript is clear, relevant for the field as short bowel syndrome is something that should always be prevented. The paper is presented in a well-structured way although it could be extended with more detailed reports. It is clear that apple peel deformity is rare, so 17 years of difference from the first to the last case is expected but this could affect the difference in the surgical technique.
The results could be improved with more details. Please add more specific detail as which patients had stoma created (numbers in both groups). The same with the complications - enter more detail as which patients had what kind of complications. How were these complications treated? Although there is no statistical significance in postoperative stay, there is important difference in these two groups which should be considered in making the decision of the surgical technique. But the most important thing is that you evaluated these children with 5 years of follow up with normal intestinal functioning in APA patients. Where there any specific tests applied to all children?
References should be improved as only 10 out of 26 references are up to date.
If there are some intraoperative findings reported on the camera the use of these pictures could be added in the paper.
The manuscript is scientifically sound and reproducible.
It will be good for authors to consider prospective study with more details to correlate.
Conclusions are consistent with the evidence provided.
Comments on the Quality of English LanguageDear Authors,
The English language should be improved throughout the paper.
Author Response
Dear Authors, Thank you for your paper entitled " Small intestinal atresia: should we preserve the peel or toss it?". The manuscript is clear, relevant for the field as short bowel syndrome is something that should always be prevented. The paper is presented in a well-structured way although it could be extended with more detailed reports. It is clear that apple peel deformity is rare, so 17 years of difference from the first to the last case is expected but this could affect the difference in the surgical technique.
The results could be improved with more details. Please add more specific detail as which patients had stoma created (numbers in both groups). The same with the complications - enter more detail as which patients had what kind of complications. How were these complications treated? Although there is no statistical significance in postoperative stay, there is important difference in these two groups which should be considered in making the decision of the surgical technique.
Thank you for your comment, we added a table to better clarify the division in the two groups.
But the most important thing is that you evaluated these children with 5 years of follow up with normal intestinal functioning in APA patients. Where there any specific tests applied to all children?
A it is a retrospective study, unfortunately no uniform intestinal functioning tests were applied and therefore we could not compare the results. We considered as main outcome for funstiona intestinal normality enteral autonomy, meaning no need for parenteral nutrition with normal growth and development. In case of a future prospective evaluation work, we will consider this for sure.
References should be improved as only 10 out of 26 references are up to date.
Thank you for your comment. Although we agree that references should be up to date as a general consideration, unfortunately dealing with such a rare surgical condition implies that available literature is scarce, and we wanted to be sure to include all the available evidences.
If there are some intraoperative findings reported on the camera the use of these pictures could be added in the paper.
Thank you for this comment. We would like to add any relevant picture but, unfortunately, we are not able to provide any of those.
The manuscript is scientifically sound and reproducible.
It will be good for authors to consider prospective study with more details to correlate.
Conclusions are consistent with the evidence provided.
Thank you for this latter comments. We would consider that for sure.
Round 2
Reviewer 2 Report
Comments and Suggestions for Authors
Dear Authors,
Thank you for the revised version of your manuscript. I suggest you update your references and include more recent ones.
Author Response
Dear Reviewer, thank you for your latest review. We agreed to update more recent references.